# OpenReview forum: "Enhancing Query-Free Jailbreaks on Text-to-Image Models with Bimodal Guidance"
_ICLR.cc/2026/Conference — ICLR 2026 Conference Withdrawn Submission_

### Official Review · Reviewer_ugrD · 2025-10-28

**Soundness:** 3
**Presentation:** 3
**Contribution:** 2
**Rating:** 2
**Confidence:** 4

**Summary:**

This paper claims that existing LLM-powered query-free jailbreak methods suffer from two key issues: first, the textual modality only focuses on the safety of individual words while neglecting contextual information; second, the lack of supervision over the visual modality fails to ensure the generation of target NSFW content. To address these problems, the paper proposes Shadows, a bimodal-guided query-free jailbreak pipeline. Textual guidance leverages contextual information through "topic assistance" and "sentence expansion," while visual guidance ensures prompt-image perceptual consistency via a surrogate T2I model and the CLIP model. Experiments conducted on 16 open-source T2I models and 4 commercial APIs demonstrate that Shadows outperforms existing methods.

**Strengths:**

1. This paper identifies the key limitations of existing jailbreak methods, which encourages the research community to reflect on the actual development of this field.
2. The study conducts large-scale experiments across a diverse set of models: 16 open-source T2I models equipped with semantic-based text checkers and 4 commercial APIs (e.g., DALL-E-3, CogView-3) with built-in defenses. The results are credible and generalize well across different model types.
3. As an LLM-powered approach, it is significantly more efficient than optimization-based methods.
4. The research clearly states its purpose is to expose T2I model vulnerabilities for safety improvement rather than misuse. It places strict constraints on the use of related artifacts (code, adversarial prompts) and prohibits the generation of harmful content, aligning with academic ethics.

**Weaknesses:**

1. Though this paper recognizes the key limitations of the field, the proposed method (Shadows) only introduces two forms of guidance, textual guidance and visual guidance. Additionally, the technologies employed (e.g., LLM assistance, CLIP similarity calculation) lack novelty, thus limiting the paper’s overall contribution.
2. Against commercial APIs with robust built-in defenses (e.g., OpenAI’s DALL-E-3), the absolute attack success rate (ASR-M) remains notably low at 4%, highlighting the challenges in bypassing the most advanced current defense mechanisms.
3. The method offers limited technical and theoretical contributions, as its core designs are primarily heuristic in nature.
4. Compared to PGJ, Shadows requires extra computations for the visual guidance module (surrogate T2I image generation + CLIP similarity calculation), resulting in increased computational latency. This may restrict its application in resource-constrained scenarios.

**Questions:**

Please clarify the technical contribution of the paper.

---

> ### Author Response · Authors · 2025-11-21
>
> ## **Response to W1 & W3**
> Current LLM-based jailbreak attacks (such as PGJ) generally adopts heuristic design, but in practice they apparently achieve better performance than optimization-based methods (e.g., MMA, Ring). As stated in lines 081-085, in fact, the use of multiple local tools in a query-based (query target T2I model) setting has been explored and shown to yield good results (Yang et al., 2024d). However, no such study has been proposed in query-free settings. The key challenge lies in designing a pipeline that effectively utilizes different local tools for jailbreak without any feedback from the target T2I model.
>
> ---
>
> ## **Response to W2**
> It is correct that attacking commercial APIs with robust built-in defenses (e.g., OpenAI’s DALL-E-3) in our query-free setting is very challenging. However, the ASR 4% of our attack is already much higher than the baselines (<1%).
>
> ---
>
> ## **Response to W4**
> The trade-off between technical complexity/efficiency and performance is common. In particular, for query-free attacks, resource is not an important factor because the computations are purely on local models without any access to the target [a].
>
> ---
>
> [a] On Success and Simplicity: A Second Look at Transferable Targeted Attacks. NeurIPS 2021.

---

### Official Review · Reviewer_m8em · 2025-10-28

**Soundness:** 2
**Presentation:** 2
**Contribution:** 2
**Rating:** 4
**Confidence:** 4

**Summary:**

This paper presents Shadows, a query-free jailbreak pipeline for Text-to-Image (T2I) models designed to overcome two limitations of existing LLM-powered methods: neglecting contextual information in textual safety criteria and overlooking visual modality supervision when generating NSFW content. Shadows addresses these issues by leveraging bimodal guidance, which includes textual guidance through topic assistance and sentence expansion for holistic harmlessness, and visual guidance via prompt-image perceptual consistency using surrogate T2I and CLIP models. Large-scale experiments on 16 open-source models (including 8 unlearned models with defensive text checkers) and 4 commercial APIs demonstrate that Shadows outperforms existing query-free jailbreaks.

**Strengths:**

1. This paper is easy to follow.
2. The ideas of Topic Assistance and Sentence Expansion are intuitive and reasonable.
3. The evaluation on multiple models is comprehensive.

**Weaknesses:**

1. The technical depth of this work seems limited.
2. Repetition may increase the risk of being detected.
3. The visual guidance requires a surrogate T2I model and need queries.

**Questions:**

1. How is the performance is the adversarial prompt is not repeated?

2. Can the adapative defense--detect whether a input prompt has repetition or redundancy--successfully defend against this jailbreak attack?

3. What is the time cost of the proposed method compared with baselines, as the visual guidance need to query the surrogate T2I model?

4. The performance of the proposed method on BLIP metric is not consistently better. Is the generated images really harmful as expected?

---

> ### Author Response · Authors · 2025-11-21
>
> ## **Response to W1**
> Current LLM-based jailbreak attacks (such as PGJ) generally adopts heuristic design, but in practice they apparently achieve better performance than optimization-based methods (e.g., MMA, Ring). As stated in lines 081-085, in fact, the use of multiple local tools in a query-based (query target T2I model) setting has been explored and shown to yield good results (Yang et al., 2024d). However, no such study has been proposed in query-free settings. The key challenge lies in designing a pipeline that effectively utilizes different local tools for jailbreak without any feedback from the target T2I model.
>
> ---
>
> ## **Response to W2 & Q2**
> As far as we know, prior to this work, there is no defense specifically targeting repetition. If our paper can help stimulate the development of such defenses in the future, then this paper has already fulfilled its value.
>
> ---
>
> ## **Response to W3**
> Yes, querying surrgate model is needed, which follows the common transfer-based black-box setting. However, querying only a local surrogate is fundamentally different in difficulty from attacking with access to the target model. In the former case, the average ASR is below 10% (see Table 2), whereas with direct feedback from the target model and iterative optimization, the ASR can even approach 100% [a]. This gap is common in the classical adversarial example literature: transfer-based attacks (which queries a local surrogate) [b] and query-based attacks that can directly query the target model [c].
>
> ---
>
> ## **Response to Q1 & Q3**
> We have already provided the ablation results in Table 5 and the time cost in Table 4.
>
> ---
>
> ## **Response to Q4**
> We will clarify that the ASR metrics have already captured the harmfulness of generated images based on two discriminators (MHSC and LLaVA) with different architectures (see Section 4.1, Evaluation metrics). For the pornography category, we specifically use the dedicated pornography classifier, NudeNet (see Table 7). BLIP is not the main metric to assess the harmfulness of the generated images. However, even for the BLIP score, our method is the best in most cases (over 80%, see Table 2).
>
> ---
>
> [a] SneakyPrompt: Jailbreaking Text-to-image Generative Models
> [b] Boosting Adversarial Attacks with Momentum
> [c] Decision-based adversarial attacks: Reliable attacks against black-box machine learning models

---

### Official Review · Reviewer_pukG · 2025-10-31

**Soundness:** 1
**Presentation:** 1
**Contribution:** 1
**Rating:** 2
**Confidence:** 5

**Summary:**

This paper presents a 'query-free' pipeline for jailbreaking text-to-image systems. The method uses a large language model to rewrite and extend inputs through prompt engineering stages such as topic assistance and sentence expansion. It then relies on a surrogate text-to-image model and a text–image matching score to select final adversarial prompts. Experiments report stronger attack performance against semantic-based text filters and concept erasing methods.

**Strengths:**

* The paper studies an application setting that is important for safety: producing adversarial prompts without interacting with the target system.
* The experimental scope covers multiple text-to-image models and includes ablations on proposed modules.

**Weaknesses:**

1. Limited novelty. The main contribution is a prompt engineering pipeline that guides an LLM to produce attack prompts, with limited theoretical insight or methodological innovation. Similar LLM-driven attack procedures exist[1-5].
2. Under-specified dynamic pool. The 'dynamic pool' lacks key details, including pool size and admission/eviction rules, which reduces reproducibility.
3. Overstated 'query-free' claim. Although there is no interaction with the target text-to-image model, the pipeline relies on components tied to the target's filtering process when selecting final prompts. In practice, 'query-free' should mean no interaction with any modules of the target system (both the T2I model and the safety filters).
4. Missing baselines. Several closely related query-free and light-query methods are not compared[1-5]
5. Evaluation focuses on semantic text filters only, without studying broader defenses such as image-only filters or combined text+image filters.
6. Robustness to stronger defenses. The paper does not evaluate against stronger systems such as GuardT2I[6].
7. No efficiency accounting. The paper does not report end-to-end time per successful adversarial prompt, GPU hours, or related costs, which are critical for real-world feasibility.
8. Unclear dependence on LLM scale. Results do not show how performance changes with LLM size or model family; a scale study would clarify the method's reliance on rewriting ability.

[1] Fuzz-testing meets LLM-based agents: An automated and efficient framework for jailbreaking text-to-image generation models. S&P 2025. CCF-A

[2] Modifier unlocked Jailbreaking text-to-image models through prompts. S&P 2025. CCF-A

[3] Divide-and-conquer attack: Harnessing the power of llm to bypass the censorship of text-to-image generation model

[4] Surrogateprompt: Bypassing the safety filter of text-to-image models via substitution. ACM CCS 2024. CCF-A.

[5] ART: Automatic Red-teaming for Text-to-Image Models to Protect Benign Users. NeurIPS 2024. CCF-A.

[6] Guardt2i: Defending text-to-image models from adversarial prompts. NeurIPS 2024. CCF-A.

**Questions:**

Please address the weaknesses.

---

> ### Author Response · Authors · 2025-11-21
>
> ## **Response to W1**
> As stated in lines 081-085, in fact, the use of multiple local tools in a query-based (query target T2I model) setting has been explored and shown to yield good results (Yang et al., 2024d). However, no such study has been proposed in our query-free (now called target-agnostic) settings. The key challenge lies in designing a pipeline that effectively utilizes different local tools for jailbreak without any feedback from the target T2I model.
>
> ---
>
> ## **Response to W2**
> We have already provided such information in our submitted code (i.e., the {historical_references} variable in the ''attack.py''), which fully supports the reproducibility of our work. We would also add such information in our paper.
>
> ---
>
> ## **Response to W3**
> This is a misunderstanding. Our attack does not access any modules of the target (both the T2I model and the safety filters). Instead, we use a local surrogate for the T2I model (see discussion in Figure 7) and does not accume the type of safety filter (we test different types of filters and also commercial systems). We have put much efforts on making a “target-agnostic” setting. Even for the evaluation metrics, we use BLIP sacore instead of CLIP score because the CLIP model has been involved in our attack. To avoid potential confusion, we could also use “target-agnostic” to replace “query-free”.
>
> ---
>
> ## **Response to W4**
> The baselines [1, 5] require querying the target model, while our attack does not. Baselines [2, 4] have only released datasets without providing open-source code. The performance of baseline [3] has already been shown in [a] to be inferior to PGJ, which has already been compared in our paper.
>
> ---
>
> ## **Response to W5**
> As shown in Table 2, the success rate of attacks on semantic text filters is currently very low, averaging under 10%. Thus, studying attacks against such defenses is crucial. Moreover, because the text filter can spot the NSFW content when the prompt is just inout to the model, it provides more real-time detection than the post-process image filter. Semantic text filters are also far more widespread and actively deployed by companies like OpenAI (Moderator)., Aliyun (Qwen3Guard)., and Meta (LlamaGuard).
>
> ---
>
> ## **Response to W6**
> Actually, GuardT2I is not as strong as our considered defenses. We have tested GuardT2I in our initial experiments and found that its false positive rate is very high, and its capability is far inferior to our considered NSFW-Text-Classifier, Detoxify, and LlamaGuard. For instance, we find that even a completely harmless sentence like
> > “The development of deep learning has been very rapid and it has been applied in various fields”
>
> gets decoded by GuardT2I into something like
> > “The learning is learning is all I d. I am learning. I am learning a d. aa d. aa learning is a d. iq d. aa,”
>
> which loses important information and results in a low similarity score, leading to a false classification as a harmful sentence. This is due to an obvious flaw in its design principle: it assumes that a prompt with high similarity to the original after decoding is benign, while a prompt with low similarity is considered a jailbreak sample.
>
> ---
>
> ## **Response to W7 & W8**
> We have already provided efficiency results in Table 4 and LLM dependence results in Figure 6.
>
> ---
>
> [a] Perception-guided Jailbreak against Text-to-Image Model

---

### Official Review · Reviewer_YGHN · 2025-11-01

**Soundness:** 3
**Presentation:** 2
**Contribution:** 2
**Rating:** 4
**Confidence:** 3

**Summary:**

This paper identifies two key flaws in existing LLM-powered query-free jailbreaks for Text-to-Image (T2I) models: (1) a failure to consider textual context beyond individual words, making them easy to block by semantic text checkers, and (2) a lack of visual supervision, causing the generated images to "drift" from the original NSFW intent. The authors propose Shadows, a novel query-free pipeline using bimodal guidance, which significantly outperforms prior methods, achieving up to a 4x higher ASR against defended models like SafeGen.

**Strengths:**

1. The proposed Shadows pipeline is a novel solution that directly maps textual and visual guidance to the two identified problems .
2. The method is tested against a wide array of targets, including 16 open-source T2I models and 4 commercial APIs, demonstrating broad effectiveness.

**Weaknesses:**

1. The narrative logic in the abstract and introduction is perhaps too direct and could be refined to provide a more compelling setup or background for the problem.
2. The results discussion is underdeveloped. It currently focuses on describing the outcomes rather than providing a deep, analytical dive into the underlying reasons why these results were achieved.

**Questions:**

1. Can you elaborate on why the older, and presumably weaker, SDv1.5 model serves as a better surrogate for the visual guidance module than more modern models like SDXL, AuraFlow, or Flux, as shown in Figure 7? Is your criteria fair enough?
2. Your ablation study (Table 5) shows the repetition trick (w/o Harmless Comment Imitation) is highly effective against NSFW-Text-Classifier but not Detoxify, but why?
3. Regarding the ablation study, the text claims, "It can be observed that removing any component leads to worse performance in almost all metrics, validating the necessity of each component." However, the data appears to contradict this for specific cases. For instance, in Figure 5, in the NSFW-Text-Classifier, the 'M', the value is higher for the M=0 than for M=1. Likewise, for Detoxify, the '$N_J$', the value is higher for $N_J=0$ than for $N_J=1$. Could the authors please explain this apparent discrepancy and clarify why these components seem to degrade performance on these specific metrics, especially as this is not discussed in the text?

---

> ### Author Response · Authors · 2025-11-21
>
> ## **Response to W1 & W2**
> We will refine the narrative in the abstract and introduction to provide a more engaging and contextual background.
> Additionally, we will enhance the results discussion by providing deeper analysis of the underlying reasons for the observed outcomes, including the responses to the following questions.
>
> ---
>
> ## **Response to Q1**
> Regarding the transferability of adversarial examples, there is a common recognition that the ability of a surrogate model does not necessarily correlate with better transferability, while relatively complex models may even lead to **attack overfitting** [a]. Likewise, the capability of the surrogate T2I model does not exhibit a simple linear relationship with the overall ASR.
>
> Moreover, since our pipeline consists of multiple modules and thus forms a complex system, the **compatibility** of the surrogate T2I model is more important than its generative ability. Finally, other factors — such as the **text encoder** used by the surrogate T2I model — may introduce differences in text comprehension and affect the alignment between generated images and the CLIP-based similarity scores (in the visual guidance module), ultimately influencing the overall attack effectiveness.
>
> ---
>
> ## **Response to Q2**
> Indeed, we leverage the high sensitivity of **NSFW-Text-Classifier** to text repetition to motivate our method design (see lines 219–227 and the quantitative analysis in Table 1).
> As described there, this sensitivity may stem from **dataset bias** present in many text checkers, including NSFW-Text-Classifier (Li, 2022; Hanu & Unitary team, 2020).
>
> Furthermore, even on Detoxify, the ASR-L decreases from **40% to 28%**, which is still a considerable change.
> Given that repetition incurs **almost no computational cost** and does not affect sentence semantics, it remains a valuable technique, especially for systems like NSFW-Text-Classifier.
>
> ---
>
> ## **Response to Q3**
> Our statement uses “**almost all**”, which explicitly excludes special cases.  In Figure 5, for both Detoxify and NSFW-Text-Classifier, once the hyperparameter values exceed 1, the performance is **generally better** than at 0, despite the temporary dip observed from 0 to 1.
>
> ---
>
> **[a]** *Revisiting Transferable Adversarial Images: Systemization, Evaluation, and New Insights*. TPAMI

---

### Comment · Area_Chair_Sejr · 2025-11-25

Dear Reviewers,

The authors have responded to your reviews. Please review and provide your feedback and responses.

Best,

Your AC

---

> ### Author Response · Authors · 2025-11-28
> **Kindly asking for interactions.**
>
> Dear AC/reviewers,
>
> The rebuttal has been there for more than one week, but so far, NO response has been given.
>
> We especially notice many concerns caused by simply missing our results in the main paper.
>
> We kindly request the reviewers to give further comments.
>
> Best,
> Authors of Paper 15138

---

### Author Response · Authors · 2025-12-03
**Frustrated and Withdraw**

Dear (New) AC and reviewers,

We have been very frustrated that no reviewers have responded to our rebuttal, even though the original comments contain many factual errors and misunderstandings. Given the change in AC following the OpenReview data breach, we have decided to withdraw our paper.

Best,
Authors of Paper 15138

---

### Note · Authors · 2025-12-03

**Comment:**

Dear (New) AC and reviewers,

We have been very frustrated that no reviewers have responded to our rebuttal, even though the original comments contain many factual errors and misunderstandings. Given the change in AC following the OpenReview data breach, we have decided to withdraw our paper.

Best, Authors of Paper 15138

**Withdrawal Confirmation:**

I have read and agree with the venue's withdrawal policy on behalf of myself and my co-authors.